# Comprehensive Structure-Activity Relationship Analysis of Benzamide Derivatives as Histone Deacetylase 1 (HDAC1) Inhibitors

**DOI:** 10.3390/ijms26209970

**Published:** 2025-10-14

**Authors:** Jorge Soto-Delgado, Yeray A. Rodríguez-Núñez, Cristian Guerra, Luis Prent-Peñaloza, Mitchell Bacho

**Affiliations:** 1Departamento de Ciencias Químicas, Facultad de Ciencias Exactas Bello, Universidad Andrés Bello, Quillota 980, Viña del Mar 2531015, Valparaíso, Chile; l.prentpealoza@uandresbello.edu; 2Laboratorio de Síntesis y Reactividad de Compuestos Orgánicos, Departamento de Ciencias Químicas, Facultad de Ciencias Exactas, Universidad Andrés Bello, Republica 275, Santiago 8370146, Chile; c.guerramadera@uandresbello.edu; 3Departamento de Ciencias Biológicas y Químicas, Facultad de Medicina y Ciencia, Universidad San Sebastián, Campus Los Leones, Lota 2465, Providencia 7510085, Santiago, Chile; mbachol@docente.uss.cl

**Keywords:** 3D-QSAR/docking, histone deacetylase 1, benzamide inhibitors

## Abstract

A three-dimensional quantitative structure-activity relationship (3D-QSAR) analysis incorporating ligand-receptor docking alignment and molecular dynamic (MD) simulations was conducted to elucidate the potent inhibitory effects of a series of benzamide derivatives on histone deacetylase 1 (HDAC1). A comparison between ligand-based (LB) and receptor-based (RB) 3D-QSAR models using molecular docking alignment produced statistically significant results. Steric and electrostatic contour maps provided insights into the interactions surrounding the benzamide ring, revealing that an increase in electron density enhances inhibitory activity. Furthermore, MD simulations were employed to investigate protein-ligand interactions in greater detail, yielding outcomes consistent with those from 3D-QSAR and molecular docking studies. This integrated approach of molecular docking, 3D-QSAR, and energy decomposition analysis derived from MD simulations, provides a valuable framework for the rational design of more potent HDAC1 inhibitors, facilitating the synthesis of highly effective anti-tumor compounds based on benzamide scaffolds.

## 1. Introduction

The histone deacetylase (HDAC) family of metalloenzymes plays a critical role in epigenetic regulation of gene expression. HDACs are categorized into four distinct Zn^2+^ dependent classes, class I (HDAC1, 2, 3, and 8), class IIa (HDAC4, 5, 7, and 9), class IIb (HDAC6 and 10), and class IV (HDAC11), as well as class III (sirtuins), which function through a different mechanism. In this context, studies have observed that the overexpression of HDACs in several human cancers produces significant effects on key cellular processes, including cell cycle arrest, altered cell proliferation, modulation of apoptotic activity, and increased sensitivity to chemotherapy. These findings suggest that HDACs play a critical role in tumorigenesis and could potentially serve as valuable therapeutic targets for enhancing the efficacy of chemotherapy treatments. Therefore, HDACs represent one of the essential validated cancer targets [1,2,3,4]. Among these, class I HDACs are widely regarded as the most relevant targets for cancer treatment because inhibitors with activity against HDACs 1, 2, 3, and 8 generally show strong antiproliferative and inductive effects on apoptotic activity [5,6].

In the search for more efficient antitumor compounds, various types of HDAC1 inhibitors (HDAC1is) have been designed to achieve isoform selectivity, with a common pharmacophore that includes a zinc-binding group that coordinates zinc atoms, a linker region that blocks access to the catalytic pocket, and a cap region that interacts with residues at the entrance of the catalytic pocket [7,8,9]. There are four major structurally distinct classes of HDAC1is: hydroxamic acids, benzamides, short-chain fatty acids, and large cyclic peptides. Figure 1 shows the most characteristic structures as HDAC inhibitors.

All these inhibitors share a common pharmacophore composed of a zinc-metal binding motif (ZBM), a linker region (linker), and a surface recognition domain (Cap). In the case of benzamide derivatives, CPD-60 is a class 1 and 2 selective inhibitors with an inhibitory range of 10–50 nM, which is more than five-fold selective for HDAC1 [10,11,12], indicating that the substituents that are introduced into the internal cavity could have higher selectivity. Additionally, CPD-60 stands out for its attractive selectivity profile and has been used as a lead compound for developing new inhibitors targeting specific HDAC isoforms and complexes.

Computer-aided drug discovery and design methods play pivotal roles in the development of new bioactive compounds. Structure-based strategies have become indispensable for developing new molecules as potential inhibitors for specific targets [13,14,15]. For instance, the combination of molecular docking and three-dimensional quantitative structure-activity relationship (3D-QSAR) analyses has proven to be a valuable tool for interpreting experimental results and guiding the design of more potent drugs [16,17].

To understand the structural requirements for effective binding of benzamides to HDAC1, in this study, molecular docking was initially employed to perform a receptor-based (RB) scheme, together with a ligand-based (LB) alignment scheme. The models were then compared through a 3D-QSAR analysis. Subsequently, molecular dynamics (MDs) simulations and ligand-protein interaction analyses were conducted to elucidate selectivity modulation and to establish a structure-activity relationship for compounds active against HDAC1.

## 2. Results and Discussion

### 2.1. Docking Analysis

This section elucidates the docking results of benzamide inhibitors within the active site of the HDAC1 enzyme, focusing on the most characteristic ligand-receptor interactions for selected key inhibitor structures. In all instances, the benzamide scaffold ring binds within the hydrophobic cavity of the active site rather than extending outside the pocket, consistent with previously reported behavior for other benzamide compounds [10,11]. In this context, a comprehensive analysis was conducted on a set of molecules based on the distinct interactions of their molecular fragments, including the ZBM, linker region, and surface-recognition domain. However, the majority of the compounds in this set lacked a bulky substituent in the cap region; therefore, emphasis was placed on the linker region and the internal cavity. Figure 2 illustrates the molecular docking modes of the selected compounds within the HDAC1 binding site.

In most cases, the carbonyl group of ZBM in the benzamide substrates forms a stable interaction with Zn^2+^ in the catalytic domain, with an average distance of 2.3 Å. This observation is noteworthy because the nitrogen of benzamide forms a hydrogen bond with His140. The linker region mediates the most critical interaction by forming a hydrogen bond with Tyr303 at an average distance of 2.3 Å. Additionally, the compounds exhibited π-stacking interactions with Phe150, where the aromatic ring binds at an average distance of 4.67 Å. Hydrophobic interactions are also observed, involving Leu271 in the Cap region and Phe109 in the internal cavity. However, variations in the aromatic substituents of benzamides warrant consideration, as heterocyclic rings may induce different binding behaviors. In this context, we observed that Tyr24 and Arg34 residues form hydrogen bonds in presence of heteroatom, suggesting that the aromatic ring serves as a well-suited linker fragment for the design of new inhibitors.

### 2.2. D-QSAR Statistical Results

Two alignment strategies, ligand-based (LB1 and LB2) and receptor-based (RB), were employed to develop 3D-QSAR models. A training set of 42 compounds and a test set of 15 compounds were utilized. The statistical results of 3D-QSAR modeling for both models are summarized in Table 1. PLS analysis yielded cross-validated q^2^ (leave-one-out) values of 0.71 for both LB models and 0.72 for the RB model. The correlation coefficient (R^2^) was 0.94 across all models, indicating that over 90% of the variance was explained, with low standard deviations: 0.23 for LB models and 0.22 for the RB model. These results demonstrate the successful construction of robust and reliable 3D-QSAR models. Furthermore, the external predictive abilities of the models were further evaluated using the test set, resulting in predicted test set correlation coefficients (R^2^_test_) of 0.75 and 0.82 for the LB and RB models, respectively. The predicted pIC_50_ values for the benzamide derivatives in the training set, based on both models, are presented in Appendix A. These results demonstrate that the LB and RB models are statistically significant. Additionally, the correlations between the experimental and predicted pIC_50_ values, as shown in Figure 3A–C, indicated strong consistency between the predicted and experimental data. Notably, the selected model exhibited a good fit across the entire dataset, with particularly strong performance for the most potent compounds.

Once a valid and robust model was established, determination of the applicability domain (AD) became imperative to meet another requirement set forth by the OECD [18]. The AD is defined as the response and chemical structure space within which the model generates predictions with a specified level of reliability. Ensuring the appropriate application of any model for predicting new, untested compounds is therefore critical. Among the numerous methods available to define the AD [19], this study employed the leverage method [20,21,22] (a distance-based approach). The corresponding Williams plots (standardized residuals versus leverage) for the ligand and receptor models are presented in Figure 3D–F.

Values exceeding h* in the test set would indicate unreliable predictions due to substantial extrapolation [18,22]. Notably, Compound **38**, like other members of the training set with leverage higher than h*, exerts a significant influence on the regression model. Both compounds were accurately predicted, as evidenced by their low standardized residuals. In contrast, Compound **30** in the LB1, LB2, and RB models, as well as Compound **24** in the RB model (Figure 3B), exhibited relatively large standardized residuals, suggesting that their activity was not entirely well predicted by the model. However, these residuals remained within an acceptable range (<2.5 σ). Compliance with the applicability domain (AD) during model building and validation, as demonstrated for all compounds used, represents an additional strength of the model. Therefore, the combination of favorable validation statistics, proven robustness, and a well-defined AD renders the model reliable for predicting the inhibitory activity of benzamide.

### 2.3. Interpretation of 3D-QSAR Models

From the 3D-QSAR results, contour maps were generated to illustrate the chemical properties of compounds that influence their biological activities. However, receptor-based 3D-QSAR contour plots do not directly incorporate receptor-ligand interactions; rather, they reflect variations in activity potency related to different chemical environments and the interactions in which the ligands are involved. This combined approach provides valuable insights for the rational design and modification of new molecules based on the benzamide scaffold, considering the receptor’s chemical environment [16,17]. Figure 4 depicts the contour maps of the steric and electrostatic fields for both the models. In these maps, the steric fields highlight regions where the bulkiness of substituents in the inhibitor molecule is predicted to enhance (green) or reduce (yellow) activity. Conversely, the electrostatic map indicates areas of high electron density (negative charge) with a red contour and regions of low electron density (partial positive charge) with a blue contour. A comparison of the steric maps revealed that the structural constraints and requirements for all models were aligned within the internal cavity (Figure 4A–C). The vdW interactions around the substituent on the internal cavity showed a negative effect on the activity, being probably the most important characteristic (yellow contours V1). In addition, vdW interactions around the linker and cap scaffold of benzamide compounds showed a positive effect on the activity (green contour V3). In the RB model (Figure 4C), negative steric maps (yellow contours V1) were identified near residues such as Val19, Tyr24, Arg34, Ile35, Phe109, and Leu139, indicating that substituents larger than a phenyl group would reduce activity. However, favorable steric maps (green contour V2) in front of the aromatic ring, close to Tyr24, Arg34, and Ile35, suggest that introducing small substituents in this region could enhance the activity. Notably, the yellow contour V2 was present across all models, emphasizing its importance. These maps establish important guidelines for proposing structural modifications in the internal cavity region, suggesting that achieving a balance between both contour maps is crucial. Specifically, small hydrophobic substituents on the aromatic ring could be advantageous. Additionally, the green contour near residues Asp99 and Phe205 within the surface recognition domain indicated the potential for introducing bulkier substitutions at this position.

For all models, the electrostatic interactions field (Figure 4D–F) in the internal cavity revealed that the presence of electron-rich substituent in the aromatic ring in ortho on meta position increased the activity (red contour E1). Additionally, some electron deficiency groups on the aromatic ring in the para position might improve activity (blue contour E2). LB1 and LB2 models exhibited an increase in surface distance, presumably attributable to the alignment methodology employed in this analysis. Nonetheless, the blue and red contour maps proximal to the aromatic ring substituent in the internal cavity demonstrated complementarity and consistency across all models. In this context, two salient regions are elucidated (Figure 4F). The first region, situated near Arg34 and Phe109, is characterized by contour maps indicating that ligands incorporating aryl substituents in this area benefit from small hydrogen bond acceptors or heteroatom substitutions, which are conducive to enhancing activity.

### 2.4. MD Simulations

To generate a comprehensive understanding of the interactions between benzamide compounds and HDAC1 receptors, molecular dynamics (MDs) simulations were conducted on the six most active compounds (36, 37, 49, CPD-60, CI-994, and MS275) in their docked complexes. It is noteworthy that the MDs simulations were performed considering the structural K+ in both sites, as reported by Sixto-López et al. [23,24]. The equilibrium and stability of the MD simulations were evaluated through root mean square deviation (RMSD) of the backbone atoms in HDAC1 relative to the initial structure, computed based on MD trajectories. Appendix A illustrates RMSD of the backbone atoms for all ligand-receptor complexes computed based on MD trajectories. Figure 5 presents the RMSD for the ligand (Figure 5A) and receptor (Figure 5B) during the 100 ns simulations. The results indicate that the RMSD values for the six inhibitors remain below 1.5 Å, suggesting that all inhibitors demonstrate stability within the binding pocket of HDAC1.

To evaluate the structural flexibility of ligand-receptor interactions, the Root-Mean-Square Fluctuations (RMSF) of residues and radius of gyration (Rg), which measure the change in the compactness of the protein during the simulation, are typically utilized for the equilibrated MD trajectories. RMSF analysis of the residues is shown in Figure 5C. None of the ligand-receptor complexes exhibited fluctuations exceeding 3 Å, except for residues that interacted with the linker region. These fluctuations, however, did not compromise the stability of the ligand within the active site. Notably, the complex with Compound **36** displayed significant fluctuation proximal to the Tyr204 and Phe205 residues, extending through the loop to the Lys211 residue. This phenomenon is likely attributable to the weaker affinity of phenol-hydroxyl for zinc compared to aniline-nitrogen, resulting in reduced stability of the receptor. Figure 5D shows the Rg of HDAC1. Its narrow range of 1 Å (20–21 Å) indicates the stability of the protein throughout the simulation, with no significant conformational changes in protein structure.

The binding energy analysis for inhibitors in each system was calculated utilizing the MM-PBSA method. The binding energies of compounds **36**, **37**, **49**, **CPD-60**, **CI-994**, and **MS275** were determined to be −19.1, −18.5, −24.0, −21.2, −14.4, and −16.1 kcal/mol, respectively (Table 2). These values demonstrated a strong correlation with their biological activities and suggested that the benzamide derivatives exhibit favorable substrate characteristics for the HDAC1 receptor. The van der Waals contributions (EvdWs) were calculated as −41.1, −34.7, −34.9, −36.7, −42.0, and −43.6 kcal/mol, respectively.

On the other hand, the electrostatic energy (E_ele_) contributes significantly to the binding energy, underlining the importance of electrostatic interactions during the binding process, whereas polar contributions (E_PB_) were unfavorable, probably due to the volume of the binding site and the consequent exposure of the hydrophobic regions of the ligand to the solvent. For the studied molecules, the results suggest that complex formation is favored by intermolecular electrostatic and van der Waals interactions, as well as by the non-polar component of the solvation free energy, with non-polar solvation being particularly favorable for ligand binding. While the electrostatic energy (ΔE_ele_) reflects vacuum Coulombic interactions between atomic charges, the polar solvation contribution (ΔE_polar_) accounts for the modulation of these interactions by the aqueous environment through continuum solvent models. Consequently, the unfavorable ΔE_polar_ values observed in our systems can be attributed to the desolvation penalty associated with ligand transfer to the binding site, particularly when hydrophobic moieties are exposed to the solvent, whereas the favorable ΔE_ele_ values emphasize the relevance of direct electrostatic interactions at the binding interface. Together with van der Waals forces and the apolar solvation component, these contributions indicate that complex formation is primarily stabilized by a balance between favorable intermolecular electrostatics and hydrophobic interactions, which is consistent with the residue-based decomposition analysis (Figure 6A). However, for the molecules, the results suggest that complex formation is favored by the intermolecular electrostatic and van der Waals interactions, as well as the non-polar component of the solvation free energy repulsive term (hydrophobic/cavity) energies. In contrast, non-polar solvation free energies were favorable for ligand binding.

The most significant residues contributing to binding energy were located at the internal cavity region, including Leu139, His140, and His141. These residues exhibited the higher energy contribution among the four compounds, indicating that these residues may be essential for the inhibitory activity and crucial to the future development of new inhibitors. Additionally, residues Gly300 and Gly301 were also identified as major contributors, suggesting their importance in stabilizing the ligand within the internal cavity of the receptor. This interaction is consistent with the contour map derived from 3D-QSAR analysis.

It is noteworthy that the contribution of the residues Met30 and Arg34 interacts with the aromatic ring substituent of the principal core; this interaction is consistent with the electrostatic contour map in generating stability through hydrogen bond interactions and with the negative steric contour map, which restricts the cavity from accommodating the ligand. Moreover, residues Phe150, Cys151, Phe205, and Leu271 interacted with the linker region of the inhibitor; these interactions were in accordance with the electrostatic map contours and indicated that they were stabilized by stacking between aromatic rings in this region of the receptor. However, unfavorable contributions of residues Ala136, Gly138, and Asp155 were observed. This was attributed to the residues being part of the internal cavity, generating movement restrictions in the ligands. On the other hand, the residues bound to Zn^2+^ (Asp176, His178, and Asp264) also show an unfavorable interaction energy, possibly because zinc shows different energetic tendencies for the dissociation of Zn^2+^ in the presence and absence of solvent molecules, depending on the protein environment [25]. These interactions are depicted in Figure 6B, where the blue color represents interactions favorable to energy and the red color represents unfavorable interactions.

Subsequently, the simulation trajectory was clustered to extract representative structures using the TTClust version 4.7.2 software [26]. The cluster representatives for each trajectory, along with the number of bonds between the ligand and the receptor, are listed in Table 3. Appendix A present the first and last cluster representatives of the HDAC1-ligand complexes and the mode of interaction in the enlarged portion of the image. Hydrophobic interactions emerged as the predominant interaction type in all cases, as demonstrated by the overlay of the cluster analysis and the most populated cluster. The results indicated that the aromatic rings in the linker region of all compounds were favored for receptor interactions, particularly hydrophobic interactions with Phe150 and Phe205, as well as hydrogen bonds with His140, His141, Gly149, and Gly256. It is noteworthy that the residual decomposition and cluster analyses provide a comprehensive understanding of the interactions that govern the affinity of these systems in HDAC1.

Therefore, energy decomposition analysis, in conjunction with contour map analysis, provides an important tool for developing a structural-activity model. Figure 7 summarizes the main requirements for new benzamide derivatives for high inhibition of HDAC1.

## 3. Materials and Methods

### 3.1. Dataset and Biological Activity

A total of 57 benzamide derivatives, identified as potent HDAC1 inhibitors, were obtained from the literature and utilized as the dataset for molecular modeling [10,11]. Figure 1 shows the family of the benzamide derivative scaffolds used in this study. The reported IC_50_values for HDAC1 inhibition were converted into their corresponding pIC_50_ values (−logIC_50_), which were used as dependent variables in the QSAR analyses. Appendix A provides an overview of all the structures, their metabolism values against the HDAC1 receptor, and their assignment to the training or test set.

### 3.2. Alignment Methods and 3D-QSAR Models

The molecular alignment of compounds is crucial for developing 3D-QSAR models [27]. Previously, we reported a combination of receptor-based methods and 3D-QSAR, which has been successfully applied to the design and understanding of bioactive compounds [16,17]. In this study, two different alignment criteria were investigated to derive the most robust 3D-QSAR statistical models and compare the LB and RB analyses.

For the LB model, two conformations were evaluated, due to the torsion presented by the amide bond connecting the linker to the ZMB scaffold in the compounds (model LB1, Figure 8A; model LB2, Figure 8B). Alignment was performed for each structure, through conformational sampling using quenched MDs implemented in Open3Daling verson 2.31 [28]. A conformational search was carried out on each structure, keeping the most stable conformations within an 8 kcal/mol range from the global minimum. Pairs of conformers with heavy atom Root Mean Square Deviations (RMSDs) below 0.2 Å were considered to duplicate the higher energy one that were discarded.

For 3D-QSAR analysis, the dataset was divided into a training set of 42 compounds and a test set of 15 compounds (over 25% of the total). The aligned ligand ensembles were enclosed within a grid box extending 5 Å beyond the largest molecule in each direction. The steric and electrostatic molecular interaction fields (MIFs) were computed using Open3DQSAR version 2.31 [29] with MMFF94 van der Waals (vdW) parameters and charges. Experimental values corresponding to the individual molecules were correlated with the MIF data for each alignment using partial least squares (PLS) regression. The optimal number of principal components (PCs) to extract was selected based on the best leave-one-out cross-validation performance [30], expressed as q^2^_LOO_. The predictive power of each PLS model was evaluated against the external test set and expressed as both R^2^pred and standard deviation of the error of prediction (SDEP). Along with the analysis, the Applicability Domain (AD) was determined using the leverage approach [20,21]. Leverages were obtained from the PLS regression of both models along with standardized residuals for the predicted biological activity. The results are displayed in the corresponding Williams plots [31]. The critical leverage (h*) was applied to assess whether a compound could be included within the AD, calculated as h* = 3 w/N, where w represents the total sum of leverage and N is the number of compounds.

### 3.3. Molecular Docking

Automated docking was employed to determine the optimal binding orientations and conformations of benzamide derivatives within the binding site of HDAC1. The structural data for HDAC1 used in these molecular docking simulations were obtained from X-ray crystallography, as cataloged in the RCSB Protein Data Bank (PDB ID: 4BKX) [32]. Ligand positioning was performed by comparative analysis with the binding site of HDAC2 (PDB ID: 4LY1) [33], which co-crystallized with the benzamide inhibitor CPD-60. Notably, both histone deacetylases exhibit 93% identity in multiple sequence alignments and their ligand-binding sites are highly conserved, indicating analogous binding mechanisms for benzamide inhibitors. The sequences and conserved ligand-binding sites of both the proteins are shown in Appendix A.

Molecular docking calculations were carried out using AutoDock 4.2 software [34], specifically utilizing AutoDock4Zn, a force field tailored for small molecule docking to zinc metalloproteins [35], using PyMOL version 2.5 Molecular Graphics System [36] with the AutoDock/Vina plugin [37]. The protein structure was prepared by removing water molecules and incorporating Kollman charges and polar hydrogen atoms. Residues within 6 Å of the co-crystallized ligand were designated as the binding sites for docking calculations. All molecules from the dataset were subsequently docked into the active site of HDAC1. After docking, docked poses were clustered into groups with RMS deviations lower than 1.0 Å. From the ensemble of predicted molecular complexes, the most populated cluster conformation, along with the lowest energy conformation for the most active compound docked to the receptor, was selected as the template pose for the docking alignment. Figure 8B illustrates the RB model.

Following molecular docking calculations, all protein-ligand complexes were prepared in PDB format using the Protein-Ligand Interaction Profiler (PLIP) version 2.4 software [38] and were subsequently visualized using the PyMOL version 2.5 software [36].

### 3.4. MD Simulations of the Protein-Ligand Complex

The structures derived from the molecular docking calculations were further studied through MD simulations to obtain energy data for the ligand-receptor complex and interactions of the residues involved. For the MD simulations, the best positions of the structures were selected from molecular docking results. MD simulations were performed using Amber 22 [39]. A ligand topology builder (ACPYPE) was used to generate parameters for all ligands [40,41]. GAFF2, ff19SB, and TIP3P force fields were used to describe the compounds, proteins, and water, respectively. Additional parameters for the Zn^2+^-binding site were applied to this metalloprotein [42].

Solvated complex minimization, heating to 300 K for 1 ns, density equilibration for 1 ns, and constant-pressure equilibration for 5 ns were performed. The production phase of 100 ns was run and the coordinates were recorded every 10 ps. Simulations were run with a 2 fs time step and Langevin dynamics for temperature control. MD trajectories were automatically clustered using TTClust version 4.7.2, using the elbow method to calculate the number of clusters, and a representative structure for each cluster was produced [26]. Representative structures were analyzed using PLIP version 2.4 software to determine the types and number of bonds that participated in the interaction [38].

Additionally, the interaction energy for active compounds, expressed as ΔG_bind_, was estimated using the molecular mechanics Poisson–Boltzmann surface area (MM/PBSA) method with the MMPBSA.py.MPI module implemented in Amber 22 to compare the most favorable interactions of the inhibitors [43]. Three-dimensional structures of protein-ligand complexes and MD trajectory analysis were visually inspected using the computer graphics program VMD version 1.93 [44] and PyMOL version 2.5 [36] software packages.

## 4. Conclusions

This investigation successfully integrated molecular docking and 3D-QSAR methodologies to elucidate the interaction between 57 inhibitors and the HDAC1 receptor. Three 3D-QSAR models were developed, ligand-based (LB1 and LB2) and receptor-based (RB) models, utilizing molecular docking. These models demonstrated leave-one-out cross-validation correlation coefficients (q^2^_LOO_) of 0.71 for the ligand-based models and 0.72 for the receptor model, as well as conventional correlation coefficients (R^2^) of 0.94 across all models with low standard deviations: 0.23 for the LB models and 0.22 for the RB model. Furthermore, the external predictive capabilities of these models were assessed using a test set, yielding predicted correlation coefficients (r^2^_pred_) 0.80 for the receptor model. Consequently, all models were successfully constructed.

The combination and comparison between contour maps derived from both models revealed significant interactions with key residues involved in HDAC1 inhibitory activity. Furthermore, MD simulations elucidated the detailed interactions between proteins and ligands and the results corroborated the findings obtained from 3D-QSAR models and molecular docking. In this context, the interaction energy decomposition analysis indicates that the unfavorable interaction in the internal cavity with residues Val19, Ile35, Phe109, and Leu139 forms a hydrophobic cavity where the van der Waals contribution exerts a more substantial effect on the ligand-receptor interaction. Conversely, the interaction of Tyr24, Arg34, Cys151, and Tyr303 provides information pertaining to the potential for substitution with groups or heteroatoms capable of interacting via hydrogen bond formation, which aligns with the electrostatic contour maps. Additionally, the interaction with Phe150 and Phe205 suggests that substitution with an electronegative substituent facilitates a π-stacking interaction in the linker region of the benzamide derivatives.

In conclusion, this combined strategy of molecular docking and 3D-QSAR with appropriate energy decomposition analysis offers a valuable approach for the rational design of more potent HDAC1 inhibitors. The best QSAR model obtained using receptor alignment in this establishes a foundation for the synthesis of potent antitumor compounds based on the benzamide scaffold.

## Data Availability

Data is contained within the article and Appendix A.

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
