# Peer review of "Comprehensive Structure-Activity Relationship Analysis of Benzamide Derivatives as Histone Deacetylase 1 (HDAC1) Inhibitors"

_ijms, 2025, doi:10.3390/ijms26209970_

Round 1
Reviewer 1 Report
Comments and Suggestions for Authors
The manuscript titled “Comprehensive structure–activity relationship analysis of benzamide derivatives as Histone Deacetylase 1 (HDAC1) inhibitors” by Mitchell Bacho et al. presents an integrated study combining molecular docking, 3D-QSAR modeling, and molecular dynamics (MD) simulations to explore the inhibitory activity of benzamide derivatives on HDAC1. The study is generally within the scope of the International Journal of Molecular Sciences.
However, the manuscript requires major revision before it can be considered for publication. Several issues need to be addressed regarding figure quality, clarity of structural data, the interpretation of results, and completeness of key experimental details. The following points should be carefully considered to improve the overall quality of the manuscript:
- Line 52-55, “There are four major structurally distinct classes of HDAC1i: hydroxamic acids, benzamides, and large cyclic peptides. Figure 1 shows the most characteristic structures as HDAC inhibitors” You mentioned short-chain fatty acids and cyclic peptides. While it makes sense to exclude short-chain fatty acid HDAC inhibitors due to their weak activity, it would be better to include the structure of Romidepsin in Figure 1, as it is the only FDA-approved Peptide HDAC inhibitor.
- The quality of Figure 2 is too low to convey sufficient information. I noticed several different lines, but their meanings are unclear. could you clarify what each represents (e.g., hydrogen bonds, π–π stacking interactions, etc.)? Since Zn²⁺ plays a critical role in binding, it would be better to highlight it clearly. Please also indicate that the yellow structure represents the inhibitor. Additionally, ensure that the coloring scheme used to differentiate elements in the protein and the ligand is consistent.
- Please add a list of abbreviations after the conclusion.
- Please include the PDB code of the protein used in the docking study.
- 3. Molecular docking, what are the differences between HDAC1 and HDAC2? Based on Figure S8, they appear to be identical.
- As the title implies HDAC1 selectivity and HDAC1 was used throughout the analysis, including more selectivity data would make the title better reflect the study’s scope.
- While the MD simulations demonstrate overall stability (RMSD <1.5 Å), Compound 36 shows significant fluctuations near Tyr204/Phe205 (RMSF ~3 Å). Could you please: (1) clarify how these fluctuations impact the MM-PBSA binding energy reliability (given Compound 36's large SD ±7.61 kcal/mol), (2) provide Zn²⁺-phenol hydroxyl distance data to support the "weaker affinity" claim, and (3) address whether other phenol-containing compounds show similar behavior? A supplementary figure showing Zn²⁺ coordination over time would strengthen your interpretation.
Author Response
Reviewer1
The manuscript titled “Comprehensive structure–activity relationship analysis of benzamide derivatives as Histone Deacetylase 1 (HDAC1) inhibitors” by Mitchell Bacho et al. presents an integrated study combining molecular docking, 3D-QSAR modeling, and molecular dynamics (MD) simulations to explore the inhibitory activity of benzamide derivatives on HDAC1. The study is generally within the scope of the International Journal of Molecular Sciences.
However, the manuscript requires major revision before it can be considered for publication. Several issues need to be addressed regarding figure quality, clarity of structural data, the interpretation of results, and completeness of key experimental details. The following points should be carefully considered to improve the overall quality of the manuscript:
- Line 52-55, “There are four major structurally distinct classes of HDAC1i: hydroxamic acids, benzamides, and large cyclic peptides. Figure 1 shows the most characteristic structures as HDAC inhibitors” You mentioned short-chain fatty acids and cyclic peptides. While it makes sense to exclude short-chain fatty acid HDAC inhibitors due to their weak activity, it would be better to include the structure of Romidepsin in Figure 1, as it is the only FDA-approved Peptide HDAC inhibitor.
Agree: Figure 1 was modified according to your suggestion, incorporating what was requested and approved by the FDA, while maintaining the types of benzamide.
- The quality of Figure 2 is too low to convey sufficient information. I noticed several different lines, but their meanings are unclear. could you clarify what each represents (e.g., hydrogen bonds, π–π stacking interactions, etc.)? Since Zn²⁺ plays a critical role in binding, it would be better to highlight it clearly. Please also indicate that the yellow structure represents the inhibitor. Additionally, ensure that the coloring scheme used to differentiate elements in the protein and the ligand is consistent.
Agree: Figure 2 was changed according to your suggestion, representing the most important interactions for a better understanding of this
- Please add a list of abbreviations after the conclusion.
Agree: A list of abbreviations has been added.
- Please include the PDB code of the protein used in the docking study.
Answer: The code of the PDB structure used is found on line 356 of page 12 of the manuscript (PDBcode: 4BKX). In addition, in the refinement process (filling in the missing residues) of the structure to be used in the molecular docking, it was completed using Modeller10.5 with the sequence #Q13547 obtained from the Uniprot database and the corresponding minimization with the standard work protocol for this type of development.
- Molecular docking, what are the differences between HDAC1 and HDAC2? Based on Figure S8, they appear to be identical.
Answer: The important thing about sequence analysis is that since there is no co-crystal for the interaction of benzamide-type compounds with HDAC1, it is necessary to perform a sequence alignment, since this allows to determine how consistent the inhibition site of these benzamide-type compounds is with the co-crystal of HDAC2, in addition to the 86.4% identity that both histones present allows to have the clarity that the benzamide ligand interacts in that place. It should be noted that this work presented is not based on carrying out a study about the selectivity between enzymes, but rather it is to generate better computer-aided design models that contain better representation with the interactions of the systems with the receptor.
- As the title implies HDAC1 selectivity and HDAC1 was used throughout the analysis, including more selectivity data would make the title better reflect the study’s scope.
Answer: In this article, we do not address the issue of selectivity among HDAC enzymes, although there is data suggesting that the systems under study may exhibit it. This article is methodological in nature and presents a comparison of 3D-QSAR models generated using different alignment methods. These results show that the model based on receptor interactions is valid and consistent with a qualitative interpretation of potential receptor interactions. This allows the receptor to be used as an alignment method at the inhibitory active site. However, as a research group, we are committed to conducting studies of this type, such as those reported in, for example: (i) F. F. Wagner, et al. Chem. Sci., 2015, 6, 804; (ii) Y. J. Seo, et al. ACS Chem. Neurosci., 2014, 5, 588–596;
- While the MD simulations demonstrate overall stability (RMSD <1.5 Å), Compound 36 shows significant fluctuations near Tyr204/Phe205 (RMSF ~3 Å). Could you please: (1) clarify how these fluctuations impact the MM-PBSA binding energy reliability (given Compound 36's large SD ±7.61 kcal/mol), (2) provide Zn²⁺-phenol hydroxyl distance data to support the "weaker affinity" claim, and (3) address whether other phenol-containing compounds show similar behavior? A supplementary figure showing Zn²⁺ coordination over time would strengthen your interpretation.
Reviewer 2 Report
Comments and Suggestions for Authors
The manuscript entitled “Comprehensive structure-activity relationship analysis of benzamide derivatives as Histone Deacetylase 1 (HDAC1) inhibitors” presents a computational study combining 3D-QSAR modeling, molecular docking, and molecular dynamics simulations for 57 benzamide derivatives targeting HDAC1.
Despite the work's solid methodological foundation and comprehensive approach, its novelty is somewhat limited. Most of the applied computational techniques are standard and well-documented in the literature. The manuscript would improve with more comparisons to current QSAR models and a clearer explanation of the new structural ideas suggested. Furthermore, there are some methodological details missing and minor language and technical inconsistencies that need correction.
I suggest publishing after reconsidering and answering the following concerns:
-
Please clearly state what sets your models apart from previously published HDAC1 QSAR models. Include a direct comparison in terms of predictive performance or molecular insights. Maybe make a parallel and explain why 3D-QSAR-proposed compounds are better than AI-generated compounds.
-
Please provide more detail about how you modeled the Zn²⁺ site: did you use bonded or nonbonded parameters, or did you utilize dummy atoms? This is essential in metalloprotein studies.
-
Consider including a PCA or chemical space analysis to show the diversity of the compound set used.
-
Some of your conclusions (e.g., favorable substitution regions) could be reinforced by proposing actual new compounds or virtual hits.
-
Check and correct typographical errors (e.g., “Try303” should be “Tyr303”) and harmonize terminology (ZBM vs ZMB).
-
Please consider relocating part of Table 4 to the supplementary material and adding a schematic overview of key findings (e.g., Figure 7) earlier in the manuscript.
-
Please clarify the energetic contribution breakdown in MM/PBSA—e.g., distinguish between ΔEele and ΔEpolar more consistently.
-
If possible, add analysis or discussion of model limitations—e.g., how robust are your predictions outside the training chemical space?
- Consider performing ADMET analysis for the most promising candidates—often compounds with promising activities lack pharmacokinetic or toxicological properties to be used in further research.
- If ADMET shows promising results, will you do experimental work? Make suggestions because a lot of colleagues would do experimental work after reading theory-based manuscripts but do not want to be (for lack of a better word) rude. So if you do not plan to do experimental work for some reason, mention that in the manuscript.
Author Response
Reviewer2
The manuscript entitled “Comprehensive structure-activity relationship analysis of benzamide derivatives as Histone Deacetylase 1 (HDAC1) inhibitors” presents a computational study combining 3D-QSAR modeling, molecular docking, and molecular dynamics simulations for 57 benzamide derivatives targeting HDAC1.
Despite the work's solid methodological foundation and comprehensive approach, its novelty is somewhat limited. Most of the applied computational techniques are standard and well-documented in the literature. The manuscript would improve with more comparisons to current QSAR models and a clearer explanation of the new structural ideas suggested. Furthermore, there are some methodological details missing and minor language and technical inconsistencies that need correction.
I suggest publishing after reconsidering and answering the following concerns:
- Please clearly state what sets your models apart from previously published HDAC1 QSAR models. Include a direct comparison in terms of predictive performance or molecular insights. Maybe make a parallel and explain why 3D-QSAR-proposed compounds are better than AI-generated compounds.
Agree: One of the challenges with predictive models is molecular alignment, a crucial step for obtaining a good QSAR. (N. Dessalew, D. S. Patel, and P. V. Bharatam, J. Mol. Graphics Modell. 2007, 25, 885.) In this regard, we compared different alignment methods, receptor-based and ligand-based. This result indicates that the methods that use molecular docking as the alignment method provide good QSAR statistics. However, and more importantly, we can qualitatively interpret the interactions with the receptor, on the other hand, the next step in our development is to use AI tools to automate the pose selection and receiver alignment process, thereby providing a better description of the model. Therefore, this work demonstrates that an alignment generated with Docking is useful and improves the statistical analysis of QSAR.
- Please provide more detail about how you modeled the Zn²⁺ site: did you use bonded or nonbonded parameters, or did you utilize dummy atoms? This is essential in metalloprotein studies.
Agree: The model reported by Macchiagodena, et.al, was used along with the parameters of the bridging residues of Zn. This working protocol is widely cited in literature and is also incorporated in Amber software.
Ref 42 in the text. Macchiagodena, M.; Pagliai, M.; Andreini, C.; Rosato, A.; Procacci, P. Upgrading and Validation of the AMBER Force Field for Histidine and Cysteine Zinc(II)-Binding Residues in Sites with Four Protein Ligands. J. Chem. Inf. Model. 2019, 59, 3803–3816.
- Consider including a PCA or chemical space analysis to show the diversity of the compound set used.
Answer: For this type of study, we have not considered PCA analysis because they are short dynamics; moreover, it is seen that there is convergence at 100 ns of simulation. These MDs have been carried out to analyze the interactions in residues in order to demonstrate that the molecular interactions that can be obtained from the combination of surfaces and the receptor provide more information. Therefore, we are not seeking to analyze conformational changes in the enzymatic process, as more extensive MD studies are required for that. In this context, the spaceland analysis is appropriate for analyzing large configurations that the system could have, which is not the case or the fundamental objective of the study.
- Some of your conclusions (e.g., favorable substitution regions) could be reinforced by proposing actual new compounds or virtual hits.
Agree: No structures are proposed; since the objective of this study is to validate that the methodology using docking alignment for the 3D-QSAR process, along with a molecular dynamics study associated with the energy of residue decomposition, allows for elucidation or obtaining more information for new design. A general structure-activity relationship is proposed for compounds that can inhibit HDAC1. The challenge of synthesis or synthetic proposals is part of a future line that we wish to explore in our research group.
- Check and correct typographical errors (e.g., “Try303” should be “Tyr303”) and harmonize terminology (ZBM vs ZMB).
Agree: Revised and modified in the text
- Please consider relocating part of Table 4 to the supplementary material and adding a schematic overview of key findings (e.g., Figure 7) earlier in the manuscript.
Agree: Table 4 was moved to supplementary as Table S1 and a figure was incorporated that outlines the families used.
- Please clarify the energetic contribution breakdown in MM/PBSA—e.g., distinguish between ΔEele and ΔEpolar more consistently.
Agree: Incorpored in the text the explanation of the suggest.
“While the electrostatic energy (ΔEele) reflects the vacuum Coulombic interactions between atomic charges, the polar contribution to solvation (ΔEpolar) represents the modulation of these interactions by the aqueous environment using continuum solvent models. Therefore, the unfavorable polar contributions observed in our systems can be explained by the desolvation penalty associated with ligand transfer to the binding site, particularly when hydrophobic moieties are exposed to the solvent. In contrast, the favorable ΔEele values highlight the relevance of direct electrostatic interactions at the binding interface. Together with van der Waals forces and the nonpolar component of the solvation free energy, these contributions suggest that complex formation is primarily stabilized by a balance between favorable intermolecular electrostatics and hydrophobic interactions, in agreement with residue-by-residue decomposition analysis (Figure 6A).”
- If possible, add analysis or discussion of model limitations—e.g., how robust are your predictions outside the training chemical space?
Answer: We incorporated a paragraph regarding the robustness of the model. In this regard, while the model is qualitative in the interpretation of the isosurfaces, in the section on the interpretation of the 3D-QSAR model, our future goal is to develop the ligand-receptor interaction isosurface as an analysis surface in QSAR. How robust are your predictions outside the training chemical space? The robustness is described in the Applicability Domain analysis and the associated statistics.
- Consider performing ADMET analysis for the most promising candidates—often compounds with promising activities lack pharmacokinetic or toxicological properties to be used in further research.
Answer: In this type of work, we do not consider an ADMET for the most promising candidates, because the work is focused on demonstrating that we can generate a prediction or a structure-activity relationship model by using or interpreting the interactions with the receptor. Therefore, in order not to divert the readers' attention from the methodology, we do not incorporate pharmacokinetic or toxicological properties.
- If ADMET shows promising results, will you do experimental work? Make suggestions because a lot of colleagues would do experimental work after reading theory-based manuscripts but do not want to be (for lack of a better word) rude. So if you do not plan to do experimental work for some reason, mention that in the manuscript.
Answer: Of course, we would like to develop experimental work with our developed models. In the future, we hope to collaborate or establish our synthesis laboratory. However, we believe that multidisciplinary work in medicinal chemistry is essential for the development of this area. Our contribution to simulation models is aimed at that development.
Round 2
Reviewer 1 Report
Comments and Suggestions for Authors
Authors answered all questions, this version of the manuscript can be accespted.
Reviewer 2 Report
Comments and Suggestions for Authors
The authors have answered all of my concerns. The paper is ready to be published.